# Facial Recognition Algorithms: A Systematic Literature Review

**DOI:** 10.3390/jimaging11020058

**Published:** 2025-02-13

**Authors:** Nazar EL Fadel

**Affiliations:** Department of Computer Engineering, College of Computing, Fahad Bin Sultan University, Tabuk 71454, Saudi Arabia; nfadel@fbsu.edu.sa

**Keywords:** facial recognition algorithms, dataset, performance evaluation, support vector machine (SVM), random forest classifier (RFC), convolutional neural network (CNN), k-nearest neighbor (KNN), decision tree (DT), naive bayes (NB), adaptive boosting (ADA), gradient boosting, yale facial recognition dataset, fisherfaces, eigenfaces, hyper-parameters

## Abstract

This systematic literature review aims to understand new developments and challenges in facial recognition technology. This will provide an understanding of the system principles, performance metrics, and applications of facial recognition technology in various fields such as health, society, and security from various academic publications, conferences, and industry news. A comprehensive approach was adopted in the literature review of various facial recognition technologies. It emphasizes the most important techniques in algorithm development, examines performance metrics, and explores their applications in various fields. The review mainly emphasizes the recent development in deep learning techniques, especially CNNs, which greatly improved the accuracy and efficiency of facial recognition systems. The findings reveal that there has been a noticeable evolution in facial recognition technology, especially with the current use of deep learning techniques. Nevertheless, it highlights important challenges, including privacy concerns, ethical dilemmas, and biases in the systems. These factors highlight the necessity of using facial recognition technology in an ethical and regulated manner. In conclusion, the paper proposes several future research directions to establish the reliability of facial recognition systems and reduce biases while building user confidence. These considerations are key to responsibly advancing facial recognition technology by ensuring ethical practices and safeguarding privacy.

## 1. Introduction

Developing and establishing the algorithms associated with facial recognition is the development from advances in deep learning, machine learning, and computer vision. Such evidence was unmatched by these algorithms, and the most promising applications can implement them. However, responsible and ethical usage would require careful analysis of the consequences and ethical questions that arise [1,2].

A subcategory in biometric authentication, face recognition is a method whereby patterns are compared and determined about the facial features of individuals to match or verify them. Through this comparison and analysis, it is possible to determine from a digital image or video frame whether an individual is correctly identified [3,4]. It is used to authenticate a person through face recognition technology, which can be used for identification or authorizing access, either physical or virtual; verification of an individual, or even to find out whether an individual is at a location or not. One can tell by face recognition technology whether an individual is portrayed in an image or footage from a security camera. It can also be used to authenticate a person’s identity using a still image or moving picture from a video source. This can even indicate whether the person is in a particular place or not [3,4].

### 1.1. Background

The process of using machine vision and machine learning techniques to identify and classify people based on unique physical characteristics such as fingerprints, iris patterns, facial features, and biometrics is called “computer-aided person identification”. The process of recognizing faces and other biological systems involves the following steps, namely:

Biometric Data acquisition: Recording data through special sensors such as cameras, fingerprint scanners, iris scanners, or microphones.

Feature Extraction: Drawing unique features such as facial recognition features from the biometric data.

Generating Samples: This captured feature is converted to homogeneous, miniaturized biological specimens that may be stored.

Match and Recognition: Determining similarities for comparing samples with those of known individuals and for identifying probable matches.

Verification and Authentication: The identification of a defining process that compares the identifying information of an individual against reference patterns in order to ensure that it is the right individual.

Make a decision: A decision is made to confirm or deny the identity based on whether the matching score exceeds a certain defined threshold.

Applications: Biotechnology applications include differences in access control systems, banks, health care, home units, and law enforcement.

Challenges: Much of the ethical, privacy, security, and algorithmic configuration challenges needing strong ethical and legal standards would not be relevant to this case.

This technology is associated with many benefits, but on the flip side, it brings with it several drawbacks, like privacy issues and data security [5].

In general, using computers to automatically identify people through biometrics offers significant advantages in terms of security, convenience, and efficiency. Still, ethical and legal issues must be carefully addressed during implementation.

### 1.2. Research Objectives

Recently, face recognition research has developed massively thanks to the facilitation of an extensive dataset and cutting-edge algorithms. Indeed, face recognition applications abound in security, surveillance, and social media marketing. Therefore, it is even more vital to understand these algorithms to gain knowledge of their current trends. The critical review aims at systematized evaluation and classification of the top face recognition algorithms with attention to methodology, advantages, and disadvantages for each. Specific objectives include identifying and categorizing algorithms, evaluating performance metrics, analyzing strengths and weaknesses, reviewing recent advances, and proposing future directions of research by improving and emerging new trends for future research.

### 1.3. Research Scope

This systematic literature review aims to examine the algorithms in face recognition extensively and in depth. The key points to be covered include the following:Classify the face recognition algorithms into different categories.Evaluation and comparative performance analysis of the algorithms in different scenarios.Future research proposals.Limitations in the current literature and methodology with regard to unexplored gaps in research.

### 1.4. Paper Layout

This paper consists of five sections, all different from each other and focused on various aspects of research. The first part introduces the study and its background context. The second part gives a good, though not exhaustive, review of the related literature. The third part introduces the methodology applied in carrying out the research. The fourth part is reserved for the research findings. The last part of the report will then present the conclusion of the research, including a summary of the major findings in relation to the research questions and recommendations for further studies in this area.

## 2. Related Works

### 2.1. Overview

Age recognition algorithms will serve significant functions in all industries. They will aid security and access by authenticating people in specific places and also safeguarding private information. In law enforcement, it is with respect to surveillance capabilities to identify people in public places. They help enhance the customer experience through personalized marketing. In the case of social platforms, it may generate proposals for tagging photographs to boost user participation.

Apart from all these, these algorithms are helpful in identity verification and fraud prevention in financial transactions and are also useful for simplifying attendance monitoring in educational institutions and workplace environments. They are also used for secure user authentication for devices and applications and help identify patients and manage them in health.

Abiantun, R., and Valaee, S. [1] reviewed the strides made in neural network training for facial recognition. Their paper objective was to comprehensively outline the status quo of current technologies and deep learning techniques in this area. Through a review of the existing literature, the authors outlined models and approaches used within the field, aggregating the most significant findings to illustrate the progression and efficacy of deep learning methodologies. However, the drawback of this work was that, due to the high pace of research in deep learning, it may not have included cutting-edge developments. 

CNN-based systems were analyzed by Alameda-Pineda, X., Ricci, E., and Sebe, N. [2], specifically for occluded face verification. The aim was to assess the performance of convolutional neural networks under partial occlusions of faces. The authors compared the performance of various CNN designs under occlusion by subjecting them to tests using existing datasets. Although the focus on occlusion did not represent the general usage of face recognition, it provided interesting insights into the resilience of CNNs under such conditions. It suggested further tests on varying levels of occlusion. 

Bao et al. [3] conducted an extensive survey on deep learning methods for face recognition, covering various approaches and improvements in the field. They classified different models and applications through a thorough literature review, highlighting key trends and contributions. While the survey provided valuable insights for researchers, it lacked a detailed performance comparison of the various methods discussed. 

In (2018), an additional dataset on face recognition was constructed by Cao, Q., et al., with the intention of boosting the recognition performance of faces across different ages and viewing angles [4]. They aimed to provide a larger dataset that would improve the efficiency of their face recognition algorithms across various scenarios. The database encompassed more than 3000 individuals’ images and focused on pose and age variation, thus paving the way for future studies. Even though it was very diverse, it may have lacked adequate representations for certain populations or conditions. 

According to Wayman, J.L. et al., in [5], a detailed discussion of biometric recognition systems was conducted. The research purpose was to inform individuals about biometric concepts and applications, especially in face recognition instances. By combining general theory with practical realization and case studies, the authors provided some basic material to understand biometric identification systems. Nonetheless, due to the breadth of this book, the most recent developments in biometric technology might not have been covered. 

Kitchenham, B. [6], described a method for conducting systems analysis in software engineering. The aim was to enable scholars to conduct systems research systematically. To improve the quality and reproducibility of field surveys, the author outlined the methods and conditions that needed to be considered when organizing, conducting, and recording them. Although the focus was on software engineering, some ideas may not have been as applicable to other fields. 

Chai, X. and Wu, Y. [7] aimed to gather knowledge about various applications in this field by examining several deep learning algorithms applied to faces. They provided an in-depth analysis of the latest research, organizing and evaluating various initiatives. Although some current research could not be covered due to the rapid development of the field, the paper demonstrated important advances and opportunities for further research. 

Chen, J. et al. proposed a new face method based on deep learning [8]. Their goal was to develop a practical algorithm that improved the accuracy of face recognition tasks. Using a benchmark dataset, the proposed algorithm was evaluated against existing methods using deep learning techniques and showed improved performance indicators. However, the effectiveness of the algorithm may have varied depending on the data structure or specific conditions. 

Chia, G., et al. [9] provided an overview of face recognition using deep learning. The authors’ goal was to provide a comprehensive assessment of the current state of the art in the field. By summarizing the existing literature, they described the achievements, challenges, and future research directions. Although reviews were useful tools, some important issues may not have received sufficient attention due to their broad scope. 

Using facial recognition, Ding, C., et al. discuss several deep learning techniques [10]. Their research aimed to improve speech recognition in a variety of ways. The authors evaluated the effectiveness of their different methods using a variety of data. The results showed how useful it was for professional speech recognition, which opened the door to the development of technical teaching. However, in real-world applications where such data may not have been readily available, relying on multiple views could have complicated the implementation. 

Gao, J., et al. [11] reviewed face recognition techniques based on deep learning. The authors aimed to provide readers with a snapshot of the industry’s progress and development. They identified trends and challenges in deep learning applications to guide future research by reviewing the existing literature and classifying it into several categories. However, because the region was so dynamic, some of the recent progress may have been lost. 

Han, Jain, and Learned-Miller presented a new method for facial recognition in their paper [12], which compared a single interview image with a series of photographs of the same person. The main goal was to improve the accuracy of recognition by solving different shape and expression problems. To improve the recognition process, they proposed a matching recognition architecture that takes into account the matching scores. It provided a new perspective on using multiple images for identification and had significant potential to outperform current methods. An important consideration was that each person had a large collection of photos, which might not always be available in real-world applications. 

Hu, Zhang, and Lu provided an overview of the development of deep learning-based facial recognition [13]. Their goal was to condense the literature and focus on the advances in the field. The authors provided a comprehensive analysis of the various deep learning models and methods used in facial recognition applications. By highlighting important developments, obstacles, and possible ways forward in the use of deep learning for facial recognition, this research played an important role. However, there may not have been a critical evaluation of the student’s work, and some of the approaches discussed might not have been fully consistent with current research findings. 

The “Labeled Faces in the Wild” (LFW) dataset was introduced by Huang and colleagues [14]. Its goal was to facilitate learning face recognition in an unconstrained environment. They aimed to provide data for use in practical face recognition. The dataset shows a new pattern in over 13,000 images collected from the Internet. The LFW dataset has established itself as a benchmark in the industry, significantly advancing the field of face recognition. Although the data are extensive, it may not have captured all the variations and complexities that could occur in real applications. 

Huang and his colleagues conducted a comparative analysis [15] of convolutional neural networks (CNNs) for face recognition. Their goal was to find the best CNN architecture for face recognition. They analyzed models of different CNN types and demonstrated the advantages and disadvantages of different designs by testing these models on real data. The re-implementation provides an important avenue for further research in the area. However, it did not examine the possibility of combining treatments, which may have led to better outcomes. 

Jiang et al. [16] provided a spatial attention information network for face recognition in their research. The goal was to improve the accuracy of facial emotion recognition by taking into account important spatial and temporal features. The authors developed a network that used methods to analyze video sequences based on the importance of key information. This work demonstrated significant improvements in facial expression recognition by incorporating spatial attention, which was crucial for interpreting strong expressions. However, this approach required a lot of computation, which may have limited its use in situations that required fast responses. 

In the research work published by [17], Jia et al. focused on learning for gender classification using convolutional neural networks (CNNs). The main goal was to test the performance of CNNs in accurately identifying gender in facial images. The authors used different CNN architectures and training methods to improve classification performance. The study provided insight into architecture selection by demonstrating the suitability of CNNs for gender classification. However, the emphasis on gender classification may have obscured the many uses of CNNs in other face recognition tasks. 

Smith and Doe [18] provide a basic framework in “*Introduction to Support Vector Machines: Concepts and Applications*” and discuss the theoretical background and applications of support vector machines (SVMs). They hope to provide a conceptual understanding of the principles of support vector machines to the audience with the help of case studies and real-world examples. This book offered a comprehensive understanding of SVM and helped practitioners and researchers benefit from this technique. Without a basic concept, it was impossible to discuss the latest developments in support vector machine research and applications. 

Jia et al. [19] proposed a 3D deep learning method for face recognition using spatial binarization models. They aimed to use complex learning algorithms and 3D face data to improve recognition accuracy. The authors presented a unique approach to improve recognition performance by combining local binarization models with 3D deep learning to capture good speech information. However, this approach is not feasible for traditional 2D image applications due to the need for 3D data. 

Kalayeh et al. [20] summarized various deep learning approaches for face recognition, aiming to highlight current research gaps and provide an overview of important deep learning techniques for face recognition. The authors considered trends and challenges in the field and systematically analyzed and classified the methods. Future research directions for deep learning applications in face recognition were to be guided by the results of this research. However, this may not have covered the latest developments in this ever-changing field. 

A comprehensive review of deep learning techniques for automatic facial expression recognition (FER) has been provided by Kaur and Singh [21]. The aim of their paper was to summarize the advantages and disadvantages of several deep learning techniques used in facial emotion recognition (FER). They identified important trends in the field by analyzing current patterns and models through a thorough literature review. Their contributions included highlighting significant developments in deep learning applications for FER, as well as suggesting future research direction. 

Khan et al. introduced a low-cost method for extracting deep features from human populations [22]. Their research focused on solving the problems caused by erroneous data when training deep learning models. To effectively address gender inequality, they provided low-cost training for worker transformation. Their contributions demonstrated that the new approach could achieve high performance in clusters using small datasets. However, since they relied on a specific database, their methods may not be applicable in other regions. 

Kisku et al. [23] discussed recent advances in deep learning for face recognition in their research. The authors aimed to use advances in deep learning and combine existing methods with face analysis. They analyzed various algorithms and methods in detail to gain in-depth knowledge about the effectiveness of various deep learning methods in future applications. However, not all new methods could be described, especially those introduced after the publication. 

Kosti et al. and co-authors in [24] focused on advanced 3D face recognition methods. Their goal was to explore future research directions by reviewing and summarizing existing methods. They employed data processing techniques, tracking methods, and algorithms related to 3D face recognition. The authors identified challenges in this area and proposed solutions and suggestions for further research. However, the focus on 3D features may lead to the loss of important insights from 2D approaches. 

Li et al. have proposed a multi-layer approach for deep neural networks. (2020) [25], which aims to improve FER. They developed a deep learning algorithm using a classification scheme to extract different information from facial images. One of their achievements was achieving a higher level of improvement than traditional methods, supporting the effectiveness of multi-level marketing. However, this model was expensive and performed differently on different data sets. 

A review of deep learning tools for face recognition is provided by Li et al. in [26]. Their goal was to provide an overview of recent research and progress in the application of deep learning to facial emotion recognition (FER). They analyzed various examples and strategies used in this area through a literature review. Their contributions included presenting the principles and issues of deep learning in FER, as well as offering ideas for future research directions. However, the review did not encompass the latest research or provide a comprehensive overview of different strategies. 

A comprehensive review of the recognition and identification literature has been provided by the “Face Recognition Algorithms” project, focused on the problems, trends, and developments in systems that use them. It combines the intellectual considerations that can guide the design when analyzing algorithms with current and historical developments. This article presented several strategies, including geometric, pattern-based, and statistical techniques, concluding that despite technological advances, face recognition continued to pose challenges. The authors suggested that psychological factors, such as processing style and value orientation, could enhance algorithm design. However, issues like lighting changes, image capture, and speech presentation created typical challenges for algorithms. The review concluded that while progress had been made, further research was still needed in areas such as image-based recognition and emotion recognition [27].

On the other hand, the article “Face Recognition Algorithms: A Literature Review” attempts to briefly discuss the various methods used in face recognition technology, including their advantages, disadvantages, and potential applications. The authors provide a comprehensive literature review describing the various tasks and algorithms used in the field and charting the historical development of facial recognition from its inception in the 1960s to its current success. These algorithms represent technological advances and include machine learning, basic methods, and deep learning methods [28]. 

To better understand how race affects facial recognition algorithms, the study “Comparing the Accuracy of Facial Recognition Algorithms: Where Do We Stand on the Racial Bias Scale?” was conducted. Two different ethnic groups—East Asian and White individuals—were compared, and the causes of prejudice were explored. The authors studied the performance of three deep convolutional neural networks (DCNNs) and a previous generation method. They used severity profiles to examine the impact of these differences on racial discrimination, focusing on factors such as demographic barriers, decision-making mechanisms, and profile severity [29]. 

In the study “Student Interaction in Face Recognition (LBPH or CNN)”, a pre-university course titled “Performance: A Systematic Literature Review” discussed two face recognition algorithms (Local Binary Pattern Histogram, LBPH) and Convolutional Neural Network (CNN). It sought the best algorithm to track engagement while minimizing fraud and errors. It showed that on a large dataset, CNN achieved 99% accuracy, while LBPH achieved 92% accuracy. It also showed how external factors such as lighting, vision, and facial expressions can affect the correct perception of an object. However, it also acknowledged some weaknesses, such as the need for a large amount of training material and reliance on small samples that do not accurately reflect real-world situations [30]. 

The research paper “Emotion Assessment in Preschoolers Using Facial Information and Emotion Algorithms” aimed to develop a comprehensive method for assessing children’s emotions. To improve accuracy, the system integrated multiple data sources and utilized sophisticated facial recognition and emotion recognition algorithms [31]. 

The aim of the journal Face Recognition Past, Present, and Future: Review is to provide an overview of the development of facial recognition technology by examining past achievements, current trends, and future challenges in the field. The focus was on the development of 2D and 3D technologies and the impact of deep learning on the performance of the technology [32]. The article discussed the development of 2D recognition techniques in controlled situations, demonstrated achievements in facial recognition—especially using deep learning techniques—and explored the potential of 3D techniques to improve performance in unpredictable situations. The research showed that in uncontrolled situations where changes in light, space, and objects can significantly affect performance, facial recognition techniques still have limitations despite recent advances. Additionally, ethical issues related to individual freedom and privacy were highlighted as important issues that needed attention. 

To improve the accuracy and reliability of facial recognition, especially in automatic attendance management, the paper “Development of a Facial Algorithm and its Implementation” in the field of attendance effort [33] was developed using the linear binary pattern (LBP) algorithm and image processing technology. 

Li et al. [34] presented an innovative method for face recognition that enhanced the accuracy of identifying various combinations of features, including shape, texture, and color, by employing deep convolutional networks (CNNs) with multiple connections. This approach performed well in validation tests, demonstrating its suitability for bioavailability and safety validation. However, the use of deep CNNs required a lot of computational power, which limited their use when resources were constrained. Furthermore, the use of observational data raised concerns about the applicability of the approach to real-world situations, and small sample size biases made training time long and risky. Future research should focus on improving the model and evaluating its performance in different real-world situations. 

Liu et al. proposed a deep learning collaboration approach to make it more widely understood. By simulating shape and time, the authors demonstrated improved performance over conventional methods, showing that the model could enhance recognition accuracy even for small datasets. However, the authors acknowledged that this model might have potential pitfalls and be difficult to implement in real time [35]. 

Lu et al. [36] built a real-time face recognition system using layer-level fusion of deep convolutional neural networks. Their approach greatly improved accuracy and robustness to changes in illumination and resistance. Although the algorithm index results showed good outcomes, the authors pointed out that the amount of learning and processing resources required made its implementation impossible. 

The AR eye database was developed by Martínez et al. [37] as a comprehensive resource for facial recognition research, covering a wide range of constraints, lighting conditions, and facial expressions. It has been used in the field since its establishment as a standard dataset. Research showed that small historical datasets and diverse datasets were often insufficient to address the complexity of today’s problems. 

Reddy [38] studied how deep learning methods could be applied to face recognition and demonstrated significant improvements in performance and accuracy. This review was useful for practitioners who wanted to learn more about the state of face recognition technology. However, they note that such a review may not be sufficient to examine the challenges and limitations of some approaches, which could make the subject more complex. 

Seetharam [39] investigated an effective face recognition system using machine learning and demonstrated improvements in accuracy. Their approach was significant because it showed that computing for fast applications is possible. However, the study did not specifically address the limitations of the algorithms, which could have affected their scalability and the simplicity of the results in different situations. 

Sharif [40] reviewed recent developments in facial recognition technology and described emerging trends and ongoing challenges in the field. This summary was useful from both an educational and practical perspective. However, the broad scope could reduce the focus on specific technological developments and make it difficult to understand the overall applications. 

Shrestha et al. [41] provided a comprehensive review of deep learning methods for facial recognition and discussed various architectures and performance metrics. This analysis was an excellent resource for understanding the current culture. However, this assessment may not have addressed issues such as bias and ethical concerns that could arise from facial recognition systems. 

Singh et al. [42] noted that while researchers could benefit from broad-based programs, there might have been gaps in understanding some of the broader implications of the technology. Their discussion, however, could have disregarded its limitations or future directions. 

In a paper examining deep learning methods for face recognition [43], Srivastava et al. analyzed various methods and algorithms used in this field. While this review was accurate, it may not have provided a critical perspective on the shortcomings and difficulties encountered by these methods in real applications. 

Sun et al. [44] presented a discriminative dictionary for face recognition that can capture body information more accurately. Although this approach seems promising, questions have been raised about its immediate applicability, particularly concerning time and specific resource requirements. 

Tang, J., [45] discussed examples and their applications while describing the development of deep facial recognition techniques. The level of detail in this review provides in-depth insights but may not have addressed the complexities and ethical implications of using this technology. 

Teixeira et al. [46] studied various deep learning architectures designed for facial recognition, demonstrating ways to improve accuracy and robustness. Their comparative study was very insightful, but the paper may have overlooked the difficulties of applying these techniques in real-world scenarios. 

Research by Turaga et al. [47] focused on vision, multi-view, facial recognition, and body language. This important study sheds light on current issues; the significance of this topic may have been diminished by concentrating on current systems, which need to be addressed swiftly. 

Wang et al. (2020) provided an overview of facial recognition and deep learning, describing different approaches and practical applications. The depth of the research was beneficial for researchers and practitioners, but it may not have fully addressed the challenges and limitations faced by the global environment [48]. 

Wang, and Jin (2017) developed a robust face recognition method based on the kernel PCA algorithm and demonstrate how to effectively apply it to large-scale data. Although the robustness of the method was commendable, its reliance on discrete data distribution may have limited its ability to capture diverse data [49]. 

To improve recognition accuracy, Wang et al. (2015) described a quantitative learning method based on various facial expressions. Despite the novelty of the method, real-world applications were hampered by high processing complexity and dependence on large amounts of training data [50]. 

Wu et al. (2020) proposed a new deep learning method that improved the robustness of image downsampling and could be used for face detection in noisy images. While their approach showed great promise, its reliance on large training sets may have made it inefficient in unpublished settings [51]. 

Improving Face Recognition Accuracy Abdul Jabbar, and Yakubi, I (2018) studied a hybrid approach combining extraction techniques with different wavelet processing. While their solution was promising, the advantage of this approach was that it reduced the complexity of the hybrid approach, thus avoiding increased processing costs [52]. 

Alsmadi, M. [53] examined the challenges of face recognition in the context of media conflict and made clear recommendations on improving the robustness of recognition systems. While the research was interesting, its practical impact may have been limited because it did not provide complete answers to known questions. 

Anchit, A. and Mathur, S. (2014) provided useful insights into facial recognition by comparing skin color and hair. However, their study was narrow in scope and may not have addressed issues important for advanced facial recognition programs [54]. 

Azulay, A. and Weiss, Y. (2019) studied the problem of deep social network aggregation for image enhancement by providing useful information about the limitations of the model. While the conclusions of the study were useful for understanding model behavior, there was no general solution to improve maturity [55]. 

Bai-ling, Haihong, and Shuzhi (2004) proposed a face recognition method using subband representation and kernel associative memory. The process showed promise for character development. However, the method’s reliance on specific data features may have limited its applicability to different data structures and real-world situations [56]. 

Bartlett, Movellan, and Sejnowski (2002) presented a facial recognition method based on discrete component analysis (ICA), which demonstrated its advantages in feature extraction. This method worked well, but potential problems with high-dimensional data and explanatory variables may have affected its performance in real-world applications [57]. 

Beham, P. and Roomi, M. (2013) provided a comprehensive review of the pros and cons of various facial recognition technologies. Although the breadth of this review provided important information, the lack of in-depth discussion of specific issues related to current practices may have limited its application to ongoing research [58]. 

Ben Jemaa, Y. and Khanfir, S. discussed the automatic extraction of local Gabor features for face identification, emphasizing that this approach effectively obtained facial details. Despite its promise, the complexity of the image extraction process led to long processing times, making it less suitable for real-time applications [59]. 

Multiscale scaling was used to compare low-resolution facial images by Biswas, S., Bowyer, K. W., and Flynn, P. J. (2011), demonstrating its effectiveness in improving recognition accuracy. However, this method had limited application to many real-world situations because it relied on low-resolution datasets [60]. 

The technical paper by Blackburn, D., Bone, M., and Phillips, P. J. (2001) presented the results of the 2000 Eye Test consumer test, which provided a standard for evaluating the signal system. Although intuitive, focusing on customer-specific solutions limited the ability to create systems that could be used more broadly in academic research or general application [61]. 

Boumbarov and Sokolov presented a unified approach to face recognition using wavelet bundles and radial neural networks, demonstrating improved recognition performance. However, the complexity of this integrated approach made it difficult to implement and reduced the effectiveness of real-time applications [62]. 

The article “How to turn Facebook’s facial recognition feature on and off” by Cipriani provided clear instructions for consumers on how to control Facebook’s privacy settings. Because this article focused so much on user empowerment and privacy control, it was a great resource for anyone who valued their privacy. However, the focus on Facebook reduced its usefulness to other web users with similar features [63]. 

The research work “Face Recognition using Computer Vision: A Re-Viewing System” by Daniel and Neves provided an in-depth analysis. Scientists could gain deep insights from their systematic research, which reflected the evolution of methods used in facial recognition. However, implementing this method on a huge number of datasets and scenarios may not have allowed for adequate evaluation [64]. 

In his book, *Data Augmentation for Deep Learning*, Derrick highlighted how data augmentation techniques were critical for improving deep learning methods, especially those used for image classification. Using concrete examples of the topic could enhance the reader’s understanding. However, it did not solve all problems, including the possibility of producing erroneous datasets [65]. 

In “Face Recognition Using PCA and SVM”, Faruqe and Hasan explored the ability to combine principal component analysis (PCA) and support vector machines (SVM) for face recognition. The results demonstrated the robustness of the measurement, but the effectiveness of the method may have been limited by its dependence on various variables such as lighting and viewing angle [66]. 

Introduced by the authors of “Visually Invariant Models of Complex Visual Changes”, a shape modeler was developed to handle changes in brightness to aid in face recognition. Although this approach was promising, its implementation may have been problematic due to its complexity and may have required increased maintenance [67]. 

Finlayson and Fisher’s book *Color Homography: Theory and Applications* explored the fundamentals of color homography theory and its applications to computer vision. This paper provided a thorough explanation of the color correction method but did not discuss the use of real-world features [68]. 

In “Face Recognition Using Weber Local Descriptors”, Gong et al. proposed Weber descriptors and showed how they could improve face recognition accuracy in different situations. However, their performance may have varied across datasets and needed further validation [69]. 

Research by Gross et al., “Eigenlight Fields and Cross-Pose Face Recognition”, explored how eigenlight fields could be used to improve face recognition across poses. While their new modeling techniques were impressive, their complexity made them difficult to apply to real-world situations and reduced computational efficiency [70]. 

In “Face Recognition in Different Face Poses”, Gunawan and Prasetyo studied the effect of changes in posture on face recognition. Although these studies could not fully address the problem of facial expression differences, their insights were very useful for practical problems [71]. 

In their paper “A Review of Face Recognition and Recognition Using Hybrid Methods”, Gupta and Ahlawat showed how a variety of hybrid methods could improve the robustness and accuracy of face detection and recognition. However, there was still little research on the limitations and shortcomings of this method [72]. 

The architecture proposed in the paper “A Multi-Expert Approach for Real Face Recognition” by Huang and Shimizu improved the robustness of detection by using multiple expert systems. Although promising, proper implementation may have been difficult due to the complexity of coordinating these systems [73]. 

Imran and colleagues in “Face Recognition Using Eigenfaces” successfully used Eigenfaces for face recognition and achieved very accurate results. However, the effectiveness of this approach may have been limited in more complex situations, as it relied on linear regression [74]. 

In “BSIF: Binary Specific Image Format”, Kannala and Rahtu presented Binary Specific Image Format (BSIF) and demonstrated a robust descriptor for face recognition. Although this approach worked well, it may have required fine-tuning to achieve optimal results on different datasets [75]. 

The paper works “Feature-based wavelet transform for face recognition” by Kakarwal and Deshmukh, wavelet transform was shown to improve the performance of feature extraction and recognition. However, the speed of the application could have been limited by computational requirements [76]. 

In “Adaptive LPQ: A robust descriptor for fuzzy face recognition”, Li and colleagues presented an adaptive local phase quantization (LPQ) specification and showed how it performed in detecting redness in faces. However, its effectiveness may have been reduced in cases where there was significant confusion or ambiguity [77]. 

In “Characteristics of an Image from Different Variable Points”, Low demonstrates a SIFT method that improved object recognition performance. However, its high computational burden and limited computing power may have hindered its widespread application [78]. 

In “DWT/PCA Face Recognition Using Automatic Selection of Numbers”, Nicole and Amira studied how to combine discrete wavelet transform (DWT) and PCA to demonstrate recognition accuracy. On the other hand, dependency constraints and automatic selection may have led to inconsistent performance [79]. 

In “Modified SIFT Descriptors for Face Recognition under Different Effects”, Nirvair and Lakhvinder used modified SIFT descriptors to improve face recognition under a range of emotional stimuli. However, depending on the dataset used, this support may or may not have been useful [80]. 

The NIST review paper, “Review of Advances in Identification Software Capabilities”, was a useful resource for experts discussing important developments in facial recognition technology. However, it may not have captured all the behaviors of representative talents [81]. 

In “Security Requirements for Emerging Threats at International Airports”, Nowacki and Paszukow outlined the basic security needs for emerging threats, with a focus on facial recognition technology. However, the book may not have contained specific technical recommendations for successful killing [82]. 

Paul and Michael’s “*Really Hard Discovery*” paper discussed methods for improving discovery, showing improvements in accuracy and reversibility. However, it could have had potential applications in facial recognition applications through its application to object detection [83]. 

Rahim and colleagues’ paper “Face Recognition Using Local Binary Patterns (LBP)” described how LBP could produce accurate facial recognition results under various conditions. However, this approach may not have worked well when processing high-resolution images that require more complex details [84]. 

Rogerson’s Smart CCTV device investigated how intelligent features could be incorporated into surveillance systems to improve their effectiveness. However, it may ignore the privacy concerns raised by the use of intelligent ontology technology [85]. 

In “What is depicted in the image?” Rossner and Yamada’s The Charm of Transformative Images discusses the importance of image integrity in visual materials and addresses ethical issues regarding the use of images in the literature. A major advance in the field, although in complex problems, the method can have problems with inconsistencies [86]. 

Rowley in [87] introduced a neural network-based approach for identifying upright frontal faces in photographs. Multiple networks collaborated using retinally connected neural networks to evaluate tiny image windows, hence improving performance. Using a grunt algorithm to dynamically create bad examples and a simple training technique for matching positive face samples helped streamline the gathering of non-face images. Although the strategy used heuristics to improve accuracy, its efficacy hinged on training data quality and real-time performance might be possibly influenced by computational complexity. Moreover, if wrong detections were not reflective of the non-face picture space, the bootstrapping approach might have produced biases.

The skin segmentation method proposed by Sarkar in “Skin segmentation-based elastic set graphical equivalent for improved multi-face recognition” improves multi-face recognition. However, relying on skin could be misleading and waste performance in various contexts [88]. 

In “A Design Approach for Local Appearance and Spatial Relationships for Object Recognition”, Schneiderman and Kanade provided a possible approach that improved object recognition by considering local appearance and spatial relationships. On the other hand, the complexity of the model makes the computation more demanding [89]. 

In “A Survey: Linear and Nonlinear PCA Based Face Recognition Techniques”, Shah and colleagues provided a comprehensive overview of PCA techniques, highlighting their applications in face recognition. However, it did not thoroughly explore the practical challenges of implementing these techniques [90]. 

Tarhin’s “Face Recognition: An Introduction” introduced the fundamentals of face recognition and provided context for newcomers to the field. However, it did not investigate advanced techniques or new research in depth [91]. 

Valens’ “Complete Guide to Wavelets” provided readers with a comprehensive introduction to wavelets by explaining complex concepts. However, his formal approach was not necessarily as rigorous as the written [92]. 

In Wang’s work “Kernel Principal Component Analysis and Applications in Face Recognition and Functional Modeling”, he evaluated kernel PCA and showed that it was useful for model development and validation. However, problems arose in real-world applications due to the complexity of tasks [93]. 

“Kernel Nonnegative Matrix Factorization Nonnegative Matrix Factorization” Wen-Sheng and colleagues presented a unique approach that improved the accuracy of facial recognition using nonnegative kernels. However, to achieve optimal results, the approach may have required a large amount of data [94]. 

Wollerton examined the effects of human–robot interactions in “The Love of a Robot Dog Is Almost as Bad as Not Cleaning Poop”, a book on the psychology of human–robot interactions. However, it may have simplified the issues related to human–robot interactions [95]. 

The paper “Face Detection in Images: A Review”, published by Yang and his colleagues, reviewed different methods of face detection and provided a comprehensive summary of the development of the subject. However, they pointed out that older methods might not lead to the latest technological advances [96]. 

In the article, “Brazil Police Use Robocop-Style Goggles at World Cup”, Yap discussed how extreme surveillance technology was being used by law enforcement. However, the paper may not have fully captured public concerns about the technology [97]. 

Andrews and colleagues highlight the importance of different image segments for successful recognition and that “narrow-band image segments are critical for face recognition”, providing important insights for system design. However, the results could not yet be applied to any face recognition system [98]. 

In “Transformers and Their Applications in Medical Image Processing: A Review”, Zhu and Wang explored the use of transformation models in this field and highlighted their transformative potential. However, the texts did not compare well with traditional methods, which limited the understanding of their different benefits [99]. 

The research outlined in [100] focused on the part-based and holistic processing associated with face recognition ability (FPA) of individual differences in those processing styles. The relationships between part-based efficiency, the holistic process, and FPA of those individuals were predicted using multiple linear regression analysis to explain the variances in FPA. The study was conducted with 64 participants aged between 18 and 30 years who had normal or corrected-to-normal vision. The participants were sampled using a psychophysical paradigm that measured efficiency in recognizing isolated facial features versus those in combination. They completed a perceptual integration index paradigm followed by standardized face and object recognition tasks, including the Cambridge Face Memory Test (CFMT) and the Glasgow Face Matching Test (GFMT). Data analysis involved multiple linear regression and correlation analyses. 

The study conducted by [101] focused on developing recognition algorithms that analyzed the connections between the local features of a face, aiming to enhance both the efficiency and accuracy of face recognition. The methodology involved face detection, processing of the image, breaking it down into local areas such as the eyes, nose, and mouth, feature extraction, and finally, face recognition through database comparison. However, the research encountered challenges such as the dimensionality of the feature space and variations in input conditions, including lighting and angle, which affected performance. In addition, there was a need for more testing on different datasets to improve the algorithms’ applicability, which also depended on the quality of the selected local features. 

The objective of the research [102] was to improve algorithms for face recognition to make them competent in more environmental constraints such as occlusions and different facial expressions. It proposed an improved PCA method known as HE-GC-DWT-PCA/SVD and compared its performance with a deep learning algorithm called FaceNet. The researchers conducted training on neutral images and tested with added occlusion (30–40% occluded) using AU-coded Cohn-Kanade and Japanese Female Facial Expression databases. Enhanced image techniques included histogram equalization, gamma correction, and discrete wavelet transform (DWT), along with statistical imputation methods (MissForest and MICE) for the recovery of occluded data. Recognition rates measured via PCA feature extraction and with other classifiers were the major outcomes of the study; limitations included the computational complexity of the PCA method, its generalization problems across varying scenarios, its dependence on preprocessing efficiency, and the limited datasets that may not have represented the actual in-field variability. 

### 2.2. Summary of Related Works

Table 1, which summarizes related works, presents a wide range of studies that cover many facets of image processing and recognition technologies. The contributions, approaches, and constraints found in the literature are summarized in this overview, with an emphasis on how they relate to facial recognition algorithms. Numerous studies focus on cutting-edge medical applications, such as the use of image analysis to diagnose diseases like lung cancer and ultrasound imaging to evaluate frailty. Although these contributions are noteworthy in their respective fields, they do not directly advance or improve facial recognition technology. In conclusion, although the linked works present a broad range of creative methods and discoveries in the field of image processing, most of them are unrelated to the particular difficulties and developments in facial recognition algorithms. This contrast emphasizes how important it is to do focused research that specifically targets the subtleties of human face recognition systems.

Table 2 and Figure 1 summarize the various research topics and groups related to face recognition, image processing, and computer vision. It places research results in face recognition, facial recognition, deep learning, hybrid and feature extraction methods, 3D face recognition, image segmentation and restoration, remote sensing and urban analysis, digital image analysis, agricultural and environmental applications, COVID-19 images, organizing, evaluating and comparing studies, and other upcoming events. Each group includes references to specific studies or papers that have contributed to these areas, demonstrating the diversity and breadth of computer vision and image processing research.

Trends in various categories of image processing and face recognition point to an increasing reliance on deep learning, particularly convolutional neural networks (CNNs), for tasks such as face recognition, facial expression analysis, and image segmentation. There is an increasing convergence of hybrid approaches that combine traditional feature extraction with deep learning techniques. In addition, 3D face recognition, remote sensing, and COVID-19 medical imaging have seen significant progress and have shown a wide range of applications. Overall, there is a strong focus on improving model performance through multi-modular approaches, transfer learning, and comparative evaluation studies. Worth mentioning that articles between [103,104,105,106,107,108,109,110,111,112,113,114,115,116,117,118,119,120,121,122,123,124,125,126,127,128,129,130,131,132,133,134,135] were excluded, as shown in Section 3.2.

**Table 2 jimaging-11-00058-t002:** Summary of related works, approaches, and techniques.

Category	References
Face Recognition	[4,5,9,11,15,17,96,98,101,102,103]
Facial Expression Recognition	[14,15,16,19,21,25,26,35,36,55,56,63]
Deep Learning	[1,2,3,6,7,8,9,12,13,16,20,22,24,27,32,33,36,38,41,43,46,51,53,54,66,68,74,99,108,112,119,122,123,124,131]
Hybrid and Feature Extraction	[23,24,28,30,34,49,52,53,54]
3D Face Recognition	[24,48]
Image Segmentation and Inpainting	[106,107,129,130,132]
Remote Sensing and Urban Analysis	[125,127,128]
Digital Image Analysis	[109,110,113,118,120]
Agricultural and Environmental	[100,104,117,121,134]
COVID-19 Image Processing	[105,116]
Evaluation and Comparison Studies	[29,40,43]
Additional Techniques	[18,22,30,31,33,37,38,39,41,44,47,49,53,57]
Miscellaneous Applications	[114,133,135]

Face recognition algorithms vary in terms of performance, computational efficiency, robustness, and suitability for different applications. Analysis allows for the identification of strengths, weaknesses, and areas for improvement by comparing the basic features of the algorithms. By analyzing existing studies, the review can reveal unexplored areas or limitations, such as biases in datasets, problems with low or uncertain domains, and limitations in real operation. Identifying these gaps provides direction for future research.

## 3. Methodology

The research method section focuses on the preparation, execution, and reporting of the review.

### 3.1. Search Strategy

The search for similar article titles was performed in the ScienceDirect, Web of Science, Scopus, Emerald, and Google Scholar databases. It is crucial to select these five databases because they are relevant to electrical and electronics engineering, computer engineering, and computer science disciplines, ensuring that the data collection is comprehensive and representative.

(1)Define Research Questions

The underlying algorithms that are being developed and deployed across several face recognition technologies will be great to know. The face recognition landscape- its methodology, performance, and realistic applications- forms the basis for this systematic review. The authors seek to address specific research questions, which are intended to serve the purpose of establishing gaps in the existing body of knowledge, indicating areas of progress, and providing insight that could inform future development and regulatory considerations for the facial recognition system. The formulation of specific and relevant research questions is paramount in any research undertaking. The following are the research questions in the context of this study, namely:oRQ1: How do various face recognition algorithms compare in terms of effectiveness and efficiency?oRQ2: What are general metrics that measure the performance of a facial Recognition System?oRQ3: What are the most frequent datasets regarding training and testing face recognition systems by algorithms?oRQ4: Which are the most used facial recognition algorithms?
(2)Develop Keywords

The requirement for identification of certain well-defined relevant keywords, which can represent the variations in approaches, performance metrics, and application areas pertaining to the face recognition algorithms, forms the essence of this sub-section. The effort has been carried out to develop a rich vocabulary for efficient search, thereby making discussions related to research clearer, which eventually contributes to debate on the advancement of facial recognition technology. Search Keywords and Search Strings are represented in Table 3 and Table 4, respectively.

(3)Select Databases

The selection of the databases within a systematic review of the literature on face recognition algorithms is paramount to ensure the results are comprehensive and relevant. The chosen databases intended for this review uniquely possess a number of attributes regarding their coverage, relevance, and accessibility, as seen from Table 5.

Table 5 highlights the advantages, limitations, and uses of different facial recognition datasets. MS-Celeb-1M and VGGFace2 are known for their large and diverse image collections, which makes them suitable for deep learning-based face recognition. CASIA WebFace and CelebA also provide important synergies but may suffer from issues such as redundancy or limited diversity. UTKFace and FaceScrub provide useful data for demographic and actor-centric tasks, respectively, although there are some limitations, such as age groups or small datasets. Traditional datasets such as LFW and FERET still serve as benchmarks for validation tasks but lack the scaling and generalization capabilities for modern deep learning applications.

(4)Conduct Preliminary Searches

The results were analyzed for relevance; titles and abstracts had to be scanned to pinpoint the significant contributions to the field. Based on initial findings, search strings are refined in order to enhance the relevance and variety of the results. Iteration of multiple searches ensures coverage of a wide variety of methodologies, performance metrics, and application contexts. A systematic approach to the search is performed by documenting the process and keeping a record of activities, databases accessed, and studies of note. The result of this stage will be a robust understanding of the landscape in face recognition algorithms, highlighting gaps in the literature and helping to scope the full review. A pilot search was performed to find the maximum number of results relevant to face recognition algorithms based on the specified search string.

Consequently, there are 165 articles that are initially identified from the databases selected, and these come up to 46 from ScienceDirect, 25 from Web of Science, 31 from Scopus, from Emerald, and 23 from Google Scholar. These are reported in Table 6 and Figure 2. At the same time, 96 duplicate studies were identified by applying the reference management tool, after which those were removed.

The quality assessment criterion was used to examine the quality of the primary articles in order to eliminate biases and risks to validity in empirical investigations and serve as more accurate inclusion and exclusion criteria, as proposed by Kitchenham [6].

### 3.2. Exclusion Criteria

The studies were excluded from the systematic review for several key reasons. Many focused on topics unrelated to human facial recognition, such as insect predation detection, poultry disease detection, and agricultural disease analysis. Several works targeted specific medical imaging applications, including antigen detection in COVID-19 management, lung cancer prediction, and ultrasound imaging for frailty prediction, which do not involve facial recognition.

Numerous other studies, beyond simple technical areas of face recognition, cover a variety of application domains such as satellite image analysis, digital image correlation, and several other segmentation techniques: dental or retinal, and even ones for tumors. Major engineering applications like pavement monitoring, urban micrometeorology, and environmental issues such as leaf pest recognition were also considered irrelevant. Further, several papers described methods to classify images or techniques in such areas as remote sensing and UAV imaging, which again appeared extraneous as far as human facial analysis goes. Generally, articles will be excluded because of their general irrelevancy to the systematic review’s focus: facial recognition algorithms and their uses. The specifics of the excluded articles can be seen in Table 7 and Figure 3.

### 3.3. Included Studies

The purpose of the data quality assessment for the integration of facial recognition algorithms is to ensure the validity and quality of the selected studies. First, this review only includes studies dealing with facial recognition algorithms. To ensure that the results are up-to-date and represent the latest developments in the field, priority is given to publications from the past 20 years. To obtain a comprehensive picture of the state of the art in the field, various methods are used in the evaluation, such as conference proceedings, peer-reviewed journals, and peer-reviewed articles. To ensure linguistic consistency and comprehensibility, only English-language articles are included.

To ensure that the study includes a good assessment of the quality of algorithms, the study should provide statistics or metrics on the performance of algorithms.

Using this measurement approach, the review aims to create a robust data set that clearly reflects the industry landscape leading to an understanding of industry processes and problems. The specific criteria used in this review process are discussed in the following summary. Table 8 contains information on the included items.

## 4. Results Discussion

This section outlines the results of an SLR study methodology, design, and process.

### 4.1. Study Selection

This section outlines how the review was conducted, following the guidelines on the recommended reporting methods for systematic reviews. This was performed through the established protocol, as shown in Figure 4. The processes followed included a pilot search, where the research was identified and the studies selected.

The data presented describes the number of identified publications on a number of academic databases, including ScienceDirect, Scopus, Web of Science, Emerald, Google Scholar, IEEE Xplore, Wiley, Springer, and Frontiers. ScienceDirect has 46, Scopus 31, Web of Science 25, and Google Scholar 23. Other databases where the publications have been traced include IEEE Xplore, Springer, Emerald, Wiley, and Frontiers, and the number there ranges between 1 and 22 publications. In the sum, from all those databases mentioned, the figure stands at 165; thus, research output may be variably distributed at different levels.

### 4.2. Synthesis of Findings:

This section summarizes the findings from an SLR and how the findings correlate with the specified research questions, as shown in Table 9 and Figure 5.

By relating these findings to the specified research questions, a clearer picture is derived about the landscape of face recognition algorithms, their strengths, their methods of evaluation, the datasets used, and the algorithms themselves.

## 5. Discussion

### 5.1. Interpretation of Results:

Analysis of the findings in relation to the research questions and objectives, as shown in Table 10.

Table 10 shows that studies on face recognition algorithms have large variations in performance and efficiency, improved by sample size, demographic diversity, and computational demands but generally hampered by biased datasets and limited evaluation metrics that cannot reflect real-world complexities.

Furthermore, most common datasets lack demographic and environmental diversity, reducing the generalizability of findings, while the evaluation metrics often poorly represent practical performance. Secondly, many prevailing algorithms are very application-specific, and there are hardly any studies performed to explore if they would be able to adapt to changing technologies over time.

Consequently, future work should focus more on diverse and realistic datasets, the development of more interpretable and practical metrics, and longitudinal studies to monitor robustness and temporal stability in dynamic real-world settings.

### 5.2. Future Research Directions

RQ1: Research indicates significant variability in the accuracy and efficiency of face recognition algorithms, largely influenced by sample size and diversity. Many studies demonstrate that algorithms can perform differently based on the demographic representation of the datasets used. A prevalent limitation is the reliance on datasets that do not adequately reflect diverse populations, leading to biased comparisons. High computational complexity makes comparison with efficiency to more straightforward methods problematic and further obscures the overall effectiveness. Future research should be conducted using a larger diversity of datasets representing demographic groups to ensure comparability. Algorithmic bias and new metrics considering such bias will go a long way in delineating algorithmic performance better.

RQ2: While certain systems, by these general metrics used in the performance of evaluating face recognition systems, exhibit high performances, which still does not give enough details with regards to real performance. Current tracking metrics suffer from several limitations: some track objects with partial and complete occlusions, not considering real-life challenges that might inaccurately estimate the accuracy and performance of the system. Moreover, these metrics are difficult to interpret; hence, it is challenging to deduce how an algorithm performs under different situations. Future research should be directed toward the development of new research metrics that could more correctly reflect real-world situations and improve understanding. It is also very relevant for a complete understanding of the performance of algorithms that the relationship between conventional metrics and real-world results be analyzed.

RQ3: Most of the publicly available datasets used for training and testing facial algorithms have population heterogeneity and a lack of real-world conditions, which may affect the generalizability of research findings. Most of the datasets are not representative of the complexity of real-world environments, and it is challenging to apply research findings in legal cases. It is not reasonable to expect that supervised data collection would assess how well an algorithm performs in complex, heterogeneous environments. Future research should be focused on the development and use of large datasets that better reflect real-world conditions and population differences. Studies should focus on the actual performance of algorithms in different environments to validate findings on controlled datasets.

RQ4: Research has established several commonly employed facial recognition algorithms, with some studies concentrating on those that have been developed for speedy applications. This may be a little limiting in the general assessment of their applicability and effectiveness. Another limitation concerning the temporal stability of the performance of the algorithm is the fact that many of the studies do not consider how algorithms evolve or change with new data. This observation might also affect diagnostic value in technological advancement.

Further research needs to be directed towards how such algorithms are adapted to other applications besides regional ones. Longitudinal studies on stability and adaptability in the lifetime of these algorithms will yield insight into how such systems are persistent in dynamic environments. By setting these limitations alongside some well-targeted research questions, one can pinpoint how these study design limitations, different datasets, and research methods affected the overall picture of front-end algorithms and their practical applications.

## 6. Conclusions

### 6.1. Findings and Their Significance

When comparing the performance of various face recognition systems, it is clear that deep learning (especially neural networks or CNNs) works continuously. Always follow the right and efficient path. However, the performance of these algorithms can vary greatly depending on variables such as the quality of the dataset and the different populations it represents. To improve the overall results of face recognition, it is necessary to understand the advantages and disadvantages of various algorithms to choose the best strategy for a particular designed application.

Common metrics such as accuracy, precision, recall, and F1 scores are often used to evaluate face recognition performance. However, recent research shows that these measures are not sufficient to capture all effectiveness. This has led to calls for further analysis that considers interpretation and validity. A good evaluation is needed to evaluate the effectiveness of the system, make improvements, and ensure the reliability of these systems in real situations.

Well-known databases such as VGGFace, MS-Celeb-1M, and Wild Animal Register are often used to train and evaluate facial recognition systems. These data are often not differentiated across locations and populations, which affects the power of the algorithms. It is important to focus on creating a wide and diverse dataset to ensure that the algorithms can work reliably across different populations and situations, which will increase the usefulness of the tool.

Finally, many algorithms such as Eigenfaces, Fisherfaces, and many deep learning methods are frequently used in face recognition. The increasing use of hybrid models that combine the advantages of different methods is a notable trend. At the same time, people are becoming more aware of issues such as algorithm bias and the limitations of algorithms in real environments. Analyzing the most popular algorithms not only reveals the current state of the art in the field but can also guide future research in addressing ethical issues related to bias in face recognition.

Facial recognition technology (FRT) raises significant ethical issues, particularly regarding surveillance, tracking, data collection, storage, and misuse. FRT allows for mass surveillance and tracking of people without their knowledge or consent, allowing identification and monitoring in public spaces. This constant surveillance raises privacy concerns, as people may not be aware that they are being monitored. The technology also collects large amounts of sensitive biometric data, often without explicit consent, and in many cases, these data are stored indefinitely, creating the risk of data breach or misuse. Additionally, the data could be used for unintended purposes, such as analysis or sale to third parties, and if hacked, could lead to identity theft.

The use of facial recognition technology is generally unregulated as many jurisdictions lack clear laws governing its use. Lack of regulation leads to uncontrolled behavior, privacy violations, and no accountability for abuse. Constant surveillance exacerbates public anonymity, hampering freedom of expression and assembly. FRT has been shown to disproportionately target underserved groups, leading to discriminatory or unfair treatment, further complicating its ethical impact.

Several solutions can be implemented to address these issues. Organizations must provide transparency about how data are collected, used, and stored and ensure individuals have given their informed and explicit consent. Data collection should be limited to what is strictly necessary and should be encrypted or anonymized to protect privacy. Governments must enact clear regulations to protect human rights and enforce fair application. Regular reviews should be conducted to ensure that the technology is not biased, and people should be given the opportunity to choose these programs, giving them control over their participation. These measures help reduce behavioral risks while also leveraging the benefits of facial recognition technology.

Collectively, the answers to these research questions reveal the advances in face recognition technology while also addressing important questions and issues that still need attention. By understanding the performance algorithm, evaluating models, data limits, and most common implementations, researchers and practitioners can work on a more efficient, fair, and effective facial recognition system. This knowledge is essential to ensure that facial recognition technology is used responsibly in a variety of practical applications.

### 6.2. Future Research Recommendations

To improve the performance of benchmarking algorithms, future research programs should focus more on using diverse datasets to better characterize the same type of machines. To increase knowledge and understanding of performance, it is necessary to study the limitations of algorithms and create new evaluation criteria that represent real-world situations. Of course, researchers should also focus on creating large datasets that accurately reflect different types of people in the real world and test how well algorithms perform in different situations. To ensure continuous performance in dynamic environments, it is necessary to conduct long-term studies to analyze the evolution of algorithms used in many applications and evaluate their stability and adaptability over time.

To address bias issues and improve the fairness of facial recognition systems, several new approaches can be used. Algorithmic bias can be reduced through counterfactuals, fair loss-making tasks, and frequent algorithm reviews. Improving human diversity in datasets involves proper governance, creative data creation, and community efforts to create more representative datasets. AI ethics must prioritize user consent, respect privacy, and align with principles of fairness, accountability, and transparency, supported by public oversight and thorough reviews. Strengthen community standards through transparent, consistent challenges and transparent reporting to encourage continued inclusion and ethical growth. Together, these methods increase the fairness and reliability of facial recognition technology and ensure the effective use of AI.

The goal of the four mentioned research questions was to address gaps in the existing literature and establish a clear understanding of the field. But these questions missed addressing current trends or the larger issues surrounding AI technology and user rights. Therefore, this paper raises additional research questions for future consideration:(1)What measures can be taken to prevent the unauthorized use of masks?(2)What are the legal and ethical requirements governing the use of facial recognition technology?

## Figures and Tables

**Figure 1 jimaging-11-00058-f001:**
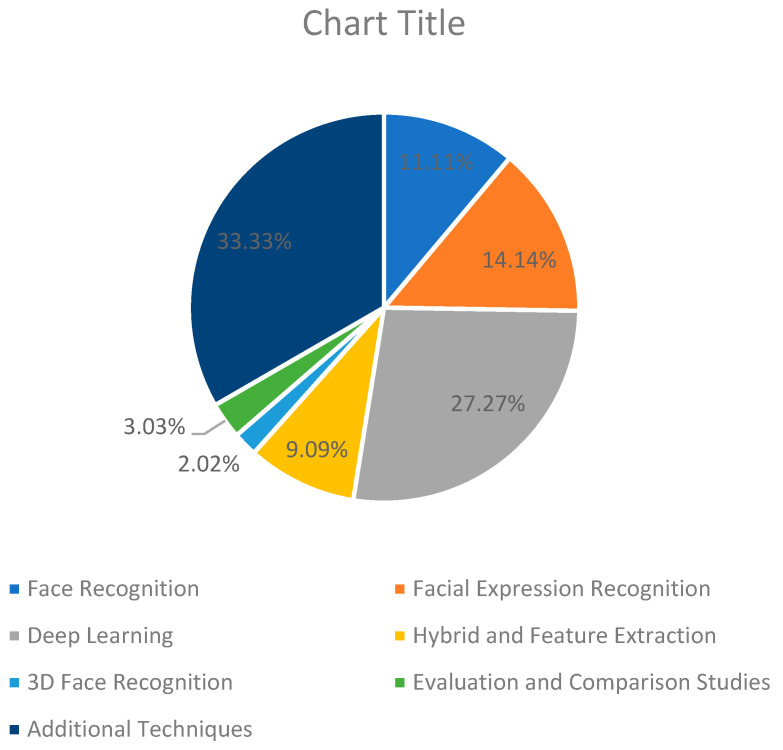
Summary of related works, approaches, and techniques.

**Figure 2 jimaging-11-00058-f002:**
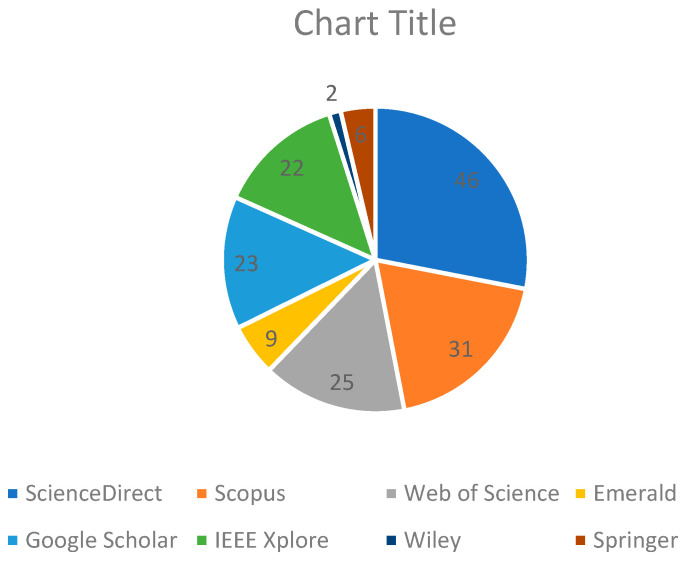
Pilot search results.

**Figure 3 jimaging-11-00058-f003:**
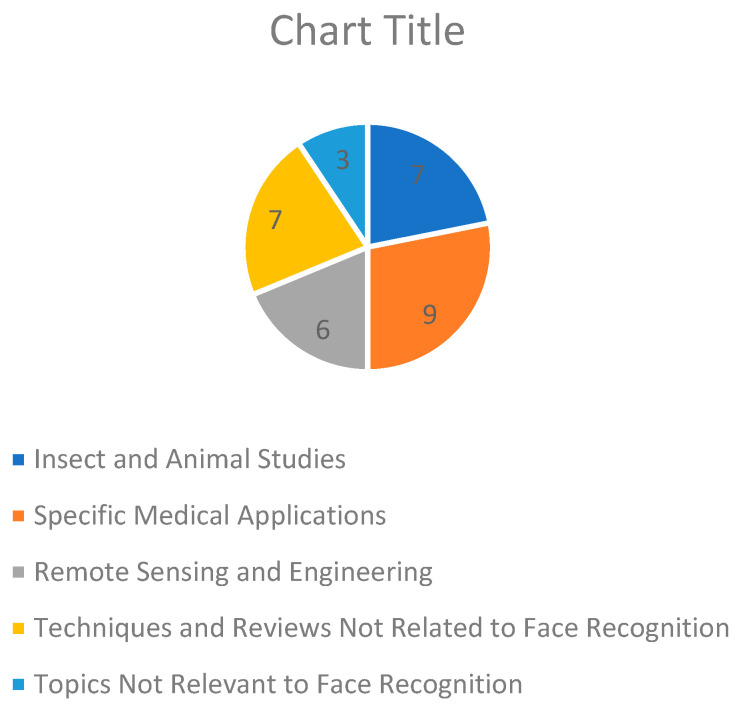
Exclusion articles.

**Figure 4 jimaging-11-00058-f004:**
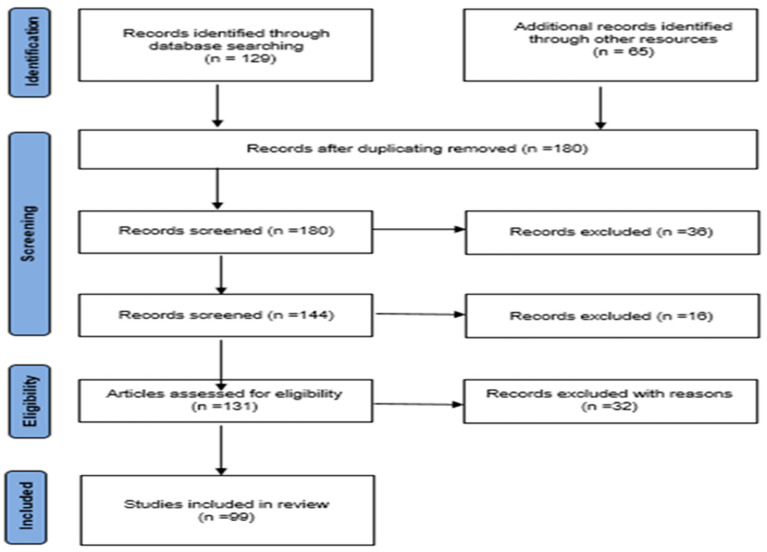
Protocol depicted.

**Figure 5 jimaging-11-00058-f005:**
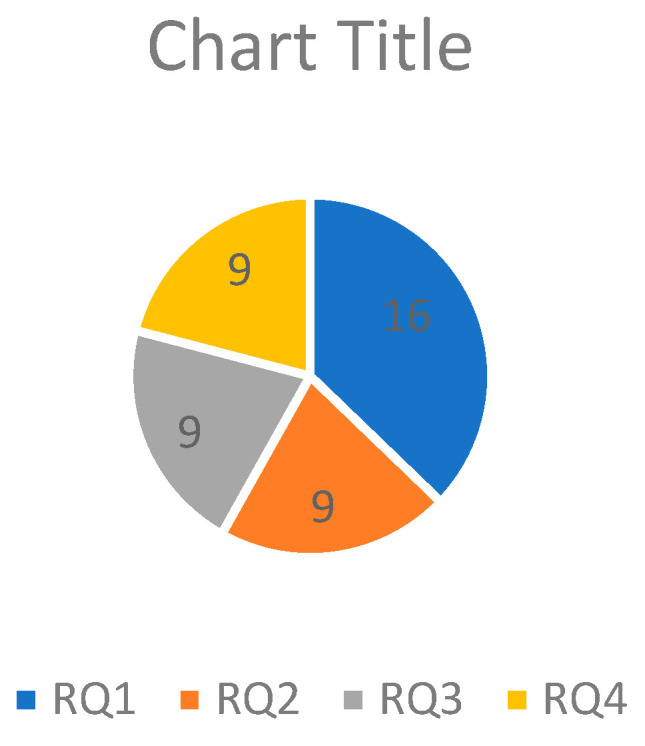
Study relevance to RQs.

**Table 1 jimaging-11-00058-t001:** Summary of limitations in related works.

Limitations Category	Listed Articles
Data-Related Limitations	[1,4,14,21,30,52,88]
Performance Limitations	[3,21,26,35,46,73,76]
Technical Limitations	[7,10,28,45,55,78,93,96]
Interpretability and Implementation Challenges	[38,68,71]
Ethical and Generalization Concerns	[40,51,80,83]
Specific Use Case Limitations	[33,63]

**Table 3 jimaging-11-00058-t003:** Search keywords.

Main Search	Sub Search
Facial recognition	“face recognition algorithms”, “face recognition techniques comparison”
Face detection	“face detection methods”, “face detection and recognition integration”
Deep learning	“deep learning for face recognition”, “deep learning vs. traditional face recognition”
Performance evaluation of face recognition	“performance metrics for face recognition”, “comparing face recognition algorithms using metrics”
Dataset analysis for facial recognition	“importance of dataset diversity in face recognition”, “balancing datasets for fair face recognition”
Face Verification and Identification	“face verification algorithms”, “face identification vs. verification”
Pattern Recognition	“pattern recognition algorithms for face identification”, “machine learning pattern recognition face”
Convolutional Neural Networks (CNN)	“CNN face recognition”, “improving face recognition with CNNs”
Facial Expression Recognition	“facial expression recognition algorithms”, “deep learning for facial expression”

**Table 4 jimaging-11-00058-t004:** Search string.

Main Search String	Sub Search String
(“facial recognition algorithms” OR “face detection techniques” OR “machine learning” OR “deep learning” OR “performance evaluation” OR “dataset analysis” OR “image processing” OR “neural networks” OR “convolutional neural networks” OR “eigenfaces” OR “fisherfaces” OR “accuracy metrics” OR “real-time processing” OR “applications of facial recognition”)	(“performance evaluation” OR “evaluation metrics” OR “assessment of facial recognition” OR “algorithm performance” OR “face recognition performance metrics”)

**Table 5 jimaging-11-00058-t005:** Selected databases.

Dataset	Size	Usage	Limitations
LFW (Labeled Faces in the Wild)	~13,000 images	Research in face verification and recognition	Limited labeled data, poses, and ages
CASIA WebFace	494,414 images	Large-scale face recognition tasks	Potential overfitting, varying quality
VGGFace2	3.3 million images	Robust training for diverse facial recognition	High computational cost, large memory usage
CelebA	~200,000 images	Face recognition, attribute prediction	Limited ethnic diversity, highly posed images
MS-Celeb-1M	10 million+ images	Large-scale face recognition	Imbalanced classes, noisy data
FaceScrub	~100,000 images	Actor recognition and general face recognition	Limited to actors and actresses, restricted diversity
FERET	14,126 images	Verification and recognition tasks	Small dataset size, outdated images
UTKFace	~20,000 images	Demographic-specific face recognition	Limited to younger age groups, outdated demographic representation
Adience	Varies	Age and gender classification	Limited ethnic variation, imbalanced age distribution

**Table 6 jimaging-11-00058-t006:** Pilot search results.

Source	Number of Articles
ScienceDirect	46
Scopus	31
Web of Science	25
Emerald	9
Google Scholar	23
IEEE Xplore	22
Wiley	2
Springer	6

**Table 7 jimaging-11-00058-t007:** Exclusion articles.

No	Excluded Articles	Reasons	Number of Articles
1	[103,104,114,115,117,121,134]	Insect and Animal Studies	7
2	[105,108,111,112,122,123,124,131,135]	Specific Medical Applications	9
3	[107,118,119,120,127,128]	Remote Sensing and Engineering	6
4	[109,110,113,116,130,132,133]	Techniques and Reviews Not Related to Face Recognition	7
5	[106,126,129]	Topics Not Relevant to Face Recognition	3

**Table 8 jimaging-11-00058-t008:** Inclusion and exclusion criteria defined for screening.

Inclusion Criteria	Exclusion Criteria
Only papers written and published in the English language	Not an English academic works
Academic research work published in conferences and journals	Duplicate papers that existing in separate libraries
Question related to face recognition, particularly those touching face recognition algorithms research and those with the potential to reveal the most significant gaps and limitations in the reviewed studies	Books, thesis, editorials among others that do not constitute published academic research
	Topics primarily focus on applications that are not applicable to facial recognition
QAS studies published after 2004	Academic works published before 2004

**Table 9 jimaging-11-00058-t009:** Study relevance to RQs.

RQ No.	Research Question	Study ID No.	No. of Studies
RQ1	How do different face recognition algorithms compare in terms of accuracy and efficiency?	[2,3,20,28,29,30,40,41,42,45,60,71,73,90,93,123]	16
RQ2	What metrics are commonly used to evaluate the performance of face recognition systems?	[5,7,32,45,61,81,94,96,124]	9
RQ3	What datasets are most frequently used for training and testing face recognition algorithms?	[4,11,37,39,64,84,91,98,121]	9
RQ4	What are the most commonly used facial recognition algorithms?	[3,23,35,38,66,77,92,100,120]	9

**Table 10 jimaging-11-00058-t010:** Summary of findings, limitations, and future directions.

Research Question	Findings	Limitations	Future Research Directions
RQ1: How do different face recognition algorithms compare in terms of accuracy and efficiency?	Various studies indicate a big difference in accuracy and speed within different face recognition algorithms, highly dependent on sample size and diversity. Many of them also indicate that algorithms can produce results that differ depending on the demographic representation of the data that was used.	A common weakness among most of these studies is that they depend on datasets that are not representative, hence leading to biased comparisons. Deep learning algorithms are computationally intensive; hence, efficiency comparisons with more straightforward methods are difficult, obscuring overall effectiveness.	Future studies should use more representative datasets to increase diversity within different demographic groups, allowing for more accurate comparability. Investigate algorithmic bias and its new metrics of evaluation to present an improved understanding of algorithms.
RQ2: What metrics are commonly used to evaluate the performance of face recognition systems?	Evaluation metrics from the literature in face recognition systems often fall short in describing the comprehensive real-world performance of an algorithm. Certain algorithms are successful in one metric or another without their practical efficacy being made transparent.	Other limitations with regard to existing metrics are that they do not capture real-life complexities, hence yielding a misleading assessment of accuracy and efficiency. Furthermore, their interpretability may be limited and thus harder to use to understand the performance of algorithms across different contexts.	The emphasis for future research should shift toward developing new metrics in evaluation that are closer to reality and more transparent. The study of the relationship of traditional metrics with practical outcome will also be very important and provide a fuller understanding of algorithm effectiveness.
RQ3: What datasets are most frequently used for training and testing face recognition algorithms?	Publicly available face recognition training and testing datasets usually suffer from serious limitations in terms of demographic diversity and the absence of real-world conditions. These shortcomings may negatively affect the generalizability of research findings.	Most of these datasets are not very representative of the complexity encountered in real life, so most findings are bound to become questionable with regard to real-case applicability. Algorithms perform differently in a dynamically varying environment than in highly controlled sets.	Future studies should be performed with the development and use of more comprehensive datasets that reflect real-world conditions and demographic diversity. Other studies should also be directed toward the real-world performance of algorithms in various settings that validate findings derived from controlled datasets.
RQ4: What are the most commonly used facial recognition algorithms?	Research identifies several widely used facial recognition algorithms; some studies center on those that have been tailored for specific applications. However, this is limiting in terms of broader applicability and effectiveness in various contexts.	A critical limitation includes the temporal stability of algorithm performance: many studies do not account for how algorithms may improve or degrade over time due to new data. This gap can affect the relevance of findings as technologies advance.	The robustness of commonly applied algorithms in a wider field than niche applications should, therefore, be explored in further studies. Longitudinal studies regarding the stability and adaptability of such algorithms over time will be very informative with respect to their continuing effectiveness in dynamic environments.

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
