# Peer review of "Facial Recognition Algorithms: A Systematic Literature Review"

_2313-433X, 2025, doi:10.3390/jimaging11020058_

Round 1

Reviewer 1 Report

Comments and Suggestions for Authors

1. The description of current mainstream algorithms is overly broad regarding technical details, advantages, and disadvantages, lacking depth and professionalism.

2. The results section lists relevant studies without providing in-depth analysis or comparison. The "Results Discussion" section significantly overlaps with the "Related Works" section, indicating a confused structure.

3. Although the paper mentions privacy and ethical concerns associated with facial recognition, it fails to provide in-depth discussion or potential solutions. The descriptions of related literature are superficial and lacking in-depth analysis or critical evaluation.

4. Although four RQs are proposed, they lack a logical progression and fail to resonate with the main body of the paper, resulting in a loose logical structure throughout the manuscript.

5. The paper primarily presents results in tables, lacking visual aids such as charts and figures, which diminishes the paper's readability.

6. There are noticeable grammatical errors, such as the one in the second line of the abstract. A complete proofreading is suggested.

Overall, this paper has certain flaws in its structure, writing, and professionalism. It also fails to track and summarize the latest research progress. The paper's structure lacks a sense of hierarchy, and its rigor needs improvement.

Author Response

Comments 1: The description of current mainstream algorithms is overly broad regarding technical details, advantages, and disadvantages, lacking depth and professionalism.

Response 1:

Thank you for your valuable feedback. In response, the author revised this section to include a more technical analysis of the algorithm, focusing on its basic methodology, performance metrics, and practical applications. Furthermore, the authors provide a balanced assessment of the pros and cons, supported by relevant empirical evidence and data. Audio visuals were added to provide a comprehensive technical overview of the current state of facial recognition algorithms. Depth analysis also provided in Table 2. 

Comments 2: The results section lists relevant studies without providing in-depth analysis or comparison. The "Results Discussion" section significantly overlaps with the "Related Works" section, indicating a confused structure.

Response 2:

Thank you for your valuable feedback. The relevant section has been reorganized to include a more in-depth analysis of Table 1 and Table 2. Additionally, the related work section order has been exchanged with the methodology section to eliminate any confusion and to establish a clearer structure. The second part offers a solid, albeit not comprehensive, overview of the pertinent literature. The third part presents the methodology employed in conducting the research.

Comments 3:  Although the paper mentions privacy and ethical concerns associated with facial recognition, it fails to provide in-depth discussion or potential solutions. The descriptions of related literature are superficial and lacking in-depth analysis or critical evaluation.

Response 3:

Thank you for your valuable feedback. A revised search strategy section has been developed to concentrate on the most pertinent elements, including the criteria for study selection, essential databases, and search terms. This section has been complemented by an updated table of selected datasets, along with illustrative figures and charts as visual aids. Additionally, the sections on related works and methodology have been rearranged in order. A comprehensive analysis, identification of gaps, and specific recommendations for future research endeavors are presented in Table 9, Section 6.1, and Section 6.2, respectively.

Comments 4:  Although four RQs are proposed, they lack a logical progression and fail to resonate with the main body of the paper, resulting in a loose logical structure throughout the manuscript.

Response 4:

Thank you for your comments. The purpose of the research questions is to address gaps in the existing literature and establish a common understanding of the field. However, the authors acknowledge that these questions may not address key developments or major challenges. Consequently, paper recommends some extra research questions that should be addressed as future recommendations; namely:

1-What measures can be implemented to prevent unauthorized use of facial data?

2- What legal frameworks are necessary to govern the ethical use of face recognition technology?

Comments 5:  The paper primarily presents results in tables, lacking visual aids such as charts and figures, which diminishes the paper's readability.

Response 5:

Thank you for your valuable comments. Audio visuals were added to improve the paper readability.

Comments 6: There are noticeable grammatical errors, such as the one in the second line of the abstract. A complete proofreading is suggested. Overall, this paper has certain flaws in its structure, writing, and professionalism. It also fails to track and summarize the latest research progress. The paper's structure lacks a sense of hierarchy, and its rigor needs improvement.

Response 6:

Thank you for your valuable feedback. The paper grammatical errors were fixed, and a comprehensive analysis, identification of gaps, and specific recommendations for future research endeavors are presented in Table 9, Section 6.1, and Section 6.2, respectively.  Additionally, the related work section has been swapped with the methodology section to eliminate any confusion and to establish a clearer structure. The second part offers a solid, albeit not comprehensive, overview of the pertinent literature. The third part presents the methodology employed in conducting the research.

Reviewer 2 Report

Comments and Suggestions for Authors

The abstract should be more concise, focusing solely on the core findings and unique contributions of the study. Currently, it includes excessive details that dilute its impact.

The introduction needs to emphasize the research gap more clearly and explicitly state why this review is necessary compared to existing literature. 

The review aims to provide a comprehensive evaluation of research trends and offer essential suggestions to the field. However, the paper fails to fulfill this role.

  • Search Strategy: A substantial portion of the paper is devoted to explaining the search strategy, which is excessive and trivial. This section could be condensed.
  • Sections 3–6: These sections lack critical comparative analysis, which is essential for a review paper.
    • Section 3.1 is a collection of summaries of related papers, without any in-depth analysis or synthesis of trends, gaps, or contradictions.
    • Section 3.2 provides only a brief summary of limitations, which is insufficient for a comprehensive review.

The five research questions presented are superficial and do not address meaningful or valuable inquiries within the field. They fail to engage with key advancements or critical challenges.

No references from 2024 are cited, which undermines the paper's currency and relevance.

The most recent advancements in facial recognition, such as the impact of generative AI, are entirely omitted. These topics are critical to understanding the current and future landscape of the field.

Ethical concerns are mentioned but not analyzed in depth. This is a crucial topic in facial recognition research and warrants a more thorough discussion.

The inclusion of diagrams or charts to represent trends, such as algorithm performance or dataset diversity, would greatly enhance the paper's readability and impact.

There are instances of awkward phrasing and grammatical inconsistencies throughout the text. Proofreading and editing for language quality and flow are strongly recommended.

Please note that a review paper plays a crucial role in academic and scientific research by synthesizing, evaluating, and contextualizing existing knowledge on a specific topic. However, the current paper focuses too much on search strategies and paper summarization. It overlooks key aspects such as evaluating trends, recommending effective methodologies and techniques, providing a critical analysis of existing works, identifying research gaps, and guiding future research.

Author Response

3. Point-by-point response to Comments and Suggestions for Authors

Comments 1: The abstract should be more concise, focusing solely on the core findings and unique contributions of the study. Currently, it includes excessive details that dilute its impact.

Response 1:

Thank you for the constructive feedback. The abstract has been revised to emphasize the primary findings and distinctive contributions of the study. Extraneous details have been eliminated to improve clarity and impact, thereby allowing the essential elements of the research to be more prominently highlighted. This modification has ensured that the abstract aligns more closely with its intended purpose of offering a concise overview for the readers.

Comments 2: The introduction needs to emphasize the research gap more clearly and explicitly state why this review is necessary compared to existing literature.

Response 2:

Thank you for the insightful feedback. The introduction has been modified to more effectively emphasize the research gap and clarify the necessity of this review. The author has made explicit comparisons between this study and the existing literature, underscoring how it addresses the unmet needs or limitations found in previous research. This improvement will offer a more robust justification for the study and elucidate its distinct contributions to the field. Details provided in section 2.3 and Table 4.

Comments 3: The review aims to provide a comprehensive evaluation of research trends and offer essential suggestions to the field. However, the paper fails to fulfill this role.

1-    Search Strategy: A substantial portion of the paper is devoted to explaining the search strategy, which is excessive and trivial. This section could be condensed.

2-    Sections 3–6: These sections lack critical comparative analysis, which is essential for a review paper.

3-    Section 3.1 is a collection of summaries of related papers, without any in-depth analysis or synthesis of trends, gaps, or contradictions.

4-    Section 3.2 provides only a brief summary of limitations, which is insufficient for a comprehensive review.

Response 3:

A revised search strategy section has been developed to concentrate on the most pertinent elements, including the criteria for study selection, essential databases, and search terms. This section has been complemented by an updated table of selected datasets, along with illustrative figures and charts as visual aids. Additionally, the sections on related works and methodology have been rearranged in order. A comprehensive analysis, identification of gaps, and specific recommendations for future research endeavors are presented in Table 9, Section 6.1, and Section 6.2, respectively.

Comments 4: The five research questions presented are superficial and do not address meaningful or valuable inquiries within the field. They fail to engage with key advancements or critical challenges.

Response 4:

Thank you for your comments. Author appreciate your concerns about the depth and relevance of our research questions. The purpose of these questions is to address gaps in the existing literature and establish a common understanding of the field. However, the authors acknowledge that these questions may not address key developments or major challenges. Consequently, paper recommends some extra research questions that should be addressed as future recommendations; namely:

1.       What measures can be implemented to prevent unauthorized use of facial data?

2.       What legal frameworks are necessary to govern the ethical use of face recognition technology?

Comments 5: No references from 2024 are cited, which undermines the paper's currency and relevance.

Response 5:

Thank you for your insightful comments. In fact, three recent articles published in 2024 have been included in the list of cited references.

5-     Audette, P.-L., Côté, L., Blais, C., Duncan, J., Gingras, F., & Fiset, D. (2025). Part-based processing, but not holistic processing, predicts individual differences in face recognition abilities. Cognition, 256, 106057. https://doi.org/10.1016/j.cognition.2024.106057

6-     Fazilova, S., Mirzaeva, O., Radjabov, S., Mirzaeva, G., & Rabbimov, I. (2024). Construction of a recognition algorithm based on the assessment of the interdependence between local elements of the face image. Procedia Computer Science, 234, 131–139. https://doi.org/10.1016/j.procs.2024.02.159.

7-     Essel, J. K., Mensah, J. A., Ocran, E., & Asiedu, L. (2024). On the search for efficient face recognition algorithm subject to multiple environmental constraints. Heliyon, 10, e28568. https://doi.org/10.1016/j.heliyon.2024.e28568

Comments 6: The most recent advancements in facial recognition, such as the impact of generative AI, are entirely omitted. These topics are critical to understanding the current and future landscape of the field.

Response 6:

Thank you for your valuable comments. The author has suggested that future developments in AI ethics should emphasize the importance of user consent, uphold privacy standards, and adhere to the principles of fairness, accountability, and transparency, all of which should be reinforced by public oversight and comprehensive evaluations.

Comments 7: Ethical concerns are mentioned but not analyzed in depth. This is a crucial topic in facial recognition research and warrants a more thorough discussion.

Response 7:

As research never ends, the current discussion has brought to light various ethical issues; however, author concur that a more thorough analysis could yield additional insights. To tackle this matter, the author has proposed it as a recommendation for future research and acknowledged it as one of the limitations of the present study.

Comments 8: The inclusion of diagrams or charts to represent trends, such as algorithm performance or dataset diversity, would greatly enhance the paper's readability and impact.

Response 8:

Thank you for your valuable comments. The author has added several charts and figures to improve the paper readability.

Comments 9: There are instances of awkward phrasing and grammatical inconsistencies throughout the text. Proofreading and editing for language quality and flow are strongly recommended.

Response 9:

Thank you for your valuable comments. Proofreading and editing for language quality and flow are done as being recommended.

Comments 10: Please note that a review paper plays a crucial role in academic and scientific research by synthesizing, evaluating, and contextualizing existing knowledge on a specific topic. However, the current paper focuses too much on search strategies and paper summarization. It overlooks key aspects such as evaluating trends, recommending effective methodologies and techniques, providing a critical analysis of existing works, identifying research gaps, and guiding future research.

Response 10:

Thank you for your valuable feedback. A comprehensive analysis, identification of gaps in current related works, and recommendations for future research were presented in Table 9, Section 6.1, and Section 6.2, respectively.

Reviewer 3 Report

Comments and Suggestions for Authors

1- The paper provides a comprehensive systematic review of facial recognition algorithms, covering their methodologies, performance metrics, and applications across various domains. It highlights advancements in deep learning, privacy concerns, and biases in existing systems. While the review is thorough and well-organized, its primary contribution lies in synthesizing existing knowledge rather than presenting novel insights. Therefore, while valuable, it is not groundbreaking.

To enhance the paper’s significance and originality, it is important to clearly distinguish this review from others in the literature, emphasizing its unique contribution. The authors should identify specific gaps in the current body of knowledge and propose innovative solutions or frameworks to address these challenges. For example, they could suggest new approaches for mitigating algorithmic biases, improving demographic diversity in datasets, or designing ethical AI practices for facial recognition systems.

2- The paper is well-structured, with clearly defined sections, including the introduction, methodology, related works, and discussion. Detailed tables, figures, and summaries effectively support the content and enhance reader comprehension. The inclusion of research questions and systematic selection criteria for reviewed studies ensures a comprehensive and methodical approach.

3- The methodology adheres to established systematic review protocols, such as Kitchenham’s guidelines, and the analysis aligns with the stated research objectives. However, the discussion of limitations could be more thorough. For instance, the paper notes issues such as a lack of demographic diversity in datasets and limitations in evaluation metrics, but these are not deeply critiqued. A more detailed analysis of dataset biases and demographic representation would strengthen the paper’s contribution and provide actionable insights for future research.

4- The paper includes an extensive and well-curated list of references to prior studies, datasets, and methods, covering a broad spectrum of the facial recognition literature. This makes it a valuable resource for readers seeking an overview of the field.

5- The English in the paper is generally clear and readable, but there are instances of awkward phrasing, particularly in the introduction and methodology sections, which would benefit from revision for clarity and flow.

I recommend

  • Simplify overly complex sentences, especially in the abstract and methodology sections.

  • Correct grammar and stylistic issues, such as redundant phrases.

In addition, correct the following errors identified in the text:

  • "Such evidence were unmatched by these algorithms, and most promising applications can realize them."

Correction: "Such evidence was unmatched by these algorithms, and the most promising applications can implement them."

  • "It carries out to authentic a person when it comes to the then face recognition technology."

Correction: "It is used to authenticate a person through face recognition technology."

  • "Nevertheless, it reveals some important issues, including privacy concerns, ethical issues, and existing biases in the systems."

Correction: "Nevertheless, it highlights important challenges, including privacy concerns, ethical dilemmas, and biases in the systems."

  • "Such considerations are key to advancing facial recognition technology development responsibly by guaranteeing ethical and privacy safeguarding creation and applications of the same."

Correction: "These considerations are key to responsibly advancing facial recognition technology by ensuring ethical practices and safeguarding privacy."

  • "By such comparison and analysis, one may ascertain via a digital image or a video frame captured from a video source whether that person is actually that person or not."

Correction: "Through this comparison and analysis, it is possible to determine from a digital image or video frame whether an individual is correctly identified."

  • "A decision is made to accept or reject the identity based on whether the matching score exceeds a set threshold."

Correction: "A decision is made to confirm or deny the identity based on whether the matching score exceeds a defined threshold."

  • "In general, using computers to automatically identify people using biometrics has significant advantages in terms of security, convenience, and efficiency, but ethical, legal, and legal issues must be carefully considered in setting up and using it."

Correction: "In general, using computers to automatically identify people through biometrics offers significant advantages in terms of security, convenience, and efficiency, but ethical and legal issues must be carefully addressed during implementation."

Author Response

Comments 1: The paper provides a comprehensive systematic review of facial recognition algorithms, covering their methodologies, performance metrics, and applications across various domains. It highlights advancements in deep learning, privacy concerns, and biases in existing systems. While the review is thorough and well-organized, its primary contribution lies in synthesizing existing knowledge rather than presenting novel insights. Therefore, while valuable, it is not groundbreaking. To enhance the paper’s significance and originality, it is important to clearly distinguish this review from others in the literature, emphasizing its unique contribution. The authors should identify specific gaps in the current body of knowledge and propose innovative solutions or frameworks to address these challenges. For example, they could suggest new approaches for mitigating algorithmic biases, improving demographic diversity in datasets, or designing ethical AI practices for facial recognition systems.

Response 1

Thank you for your insightful feedback. The analysis of gaps within the datasets has been comprehensively addressed in Table 4. Furthermore, additional information has been provided in the findings and their significance section. Detailed recommendations for future research directions have also been included in the Future Research Recommendations section.

Comments 2: The paper is well-structured, with clearly defined sections, including the introduction, methodology, related works, and discussion. Detailed tables, figures, and summaries effectively support the content and enhance reader comprehension. The inclusion of research questions and systematic selection criteria for reviewed studies ensures a comprehensive and methodical approach.

Response 2

Thank you for your insightful feedback. It is important to note that the inclusion of additional charts and figures has been implemented to enhance visual support, which may, however, reduce the overall readability of the paper.

Comments 3: The methodology adheres to established systematic review protocols, such as Kitchenham’s guidelines, and the analysis aligns with the stated research objectives. However, the discussion of limitations could be more thorough. For instance, the paper notes issues such as a lack of demographic diversity in datasets and limitations in evaluation metrics, but these are not deeply critiqued. A more detailed analysis of dataset biases and demographic representation would strengthen the paper’s contribution and provide actionable insights for future research.

Response 3

Thank you for your valuable feedback regarding the paper. Author have incorporated a more thorough analysis of biases and representation within the datasets, which will yield deeper insights. This enhancement enriches the feedback and provides actionable recommendations for future improvements in the design and application of datasets in face recognition research. These additions can be found in the Summary of Findings, Limitations, and Future Directions table (Table IX).

Comments 4: The paper includes an extensive and well-curated list of references to prior studies, datasets, and methods, covering a broad spectrum of the facial recognition literature. This makes it a valuable resource for readers seeking an overview of the field.

Response 4

Thank you for your valuable feedback and comments.

Comments 5: The English in the paper is generally clear and readable, but there are instances of awkward phrasing, particularly in the introduction and methodology sections, which would benefit from revision for clarity and flow.

I recommend

Simplify overly complex sentences, especially in the abstract and methodology sections.

Correct grammar and stylistic issues, such as redundant phrases.

In addition, correct the following errors identified in the text:

"Such evidence were unmatched by these algorithms, and most promising applications can realize them."

Correction: "Such evidence was unmatched by these algorithms, and the most promising applications can implement them."

"It carries out to authentic a person when it comes to the then face recognition technology."

Correction: "It is used to authenticate a person through face recognition technology."

"Nevertheless, it reveals some important issues, including privacy concerns, ethical issues, and existing biases in the systems."

Correction: "Nevertheless, it highlights important challenges, including privacy concerns, ethical dilemmas, and biases in the systems."

"Such considerations are key to advancing facial recognition technology development responsibly by guaranteeing ethical and privacy safeguarding creation and applications of the same."

Correction: "These considerations are key to responsibly advancing facial recognition technology by ensuring ethical practices and safeguarding privacy."

"By such comparison and analysis, one may ascertain via a digital image or a video frame captured from a video source whether that person is actually that person or not."

Correction: "Through this comparison and analysis, it is possible to determine from a digital image or video frame whether an individual is correctly identified."

"A decision is made to accept or reject the identity based on whether the matching score exceeds a set threshold."

Correction: "A decision is made to confirm or deny the identity based on whether the matching score exceeds a defined threshold."

"In general, using computers to automatically identify people using biometrics has significant advantages in terms of security, convenience, and efficiency, but ethical, legal, and legal issues must be carefully considered in setting up and using it."

Correction: "In general, using computers to automatically identify people through biometrics offers significant advantages in terms of security, convenience, and efficiency, but ethical and legal issues must be carefully addressed during implementation."

 Response 5

Thank you for your feedback on the paper. All the requested corrections were implemented and colored in yellow.

Round 2

Reviewer 1 Report

Comments and Suggestions for Authors

No more comments.

Author Response

Thank you for your valuable comments and review

Reviewer 3 Report

Comments and Suggestions for Authors

The authors have addressed all my comments correctly. The improvements are evident throughout the revised manuscript. I have no new comments. From my perspective the article can be accepted in present form.

Author Response

(The authors gave the same response as above.)
